# Development of Janus Particles as Potential Drug Delivery Systems for Diabetes Treatment and Antimicrobial Applications

**DOI:** 10.3390/pharmaceutics15020423

**Published:** 2023-01-27

**Authors:** Kei Xian Tan, Michael K. Danquah, Jaison Jeevanandam, Ahmed Barhoum

**Affiliations:** 1GenScript Biotech Pte. Ltd., 164, Kallang Way, Solaris@Kallang 164, Singapore 349248, Singapore; 2Department of Chemical Engineering, University of Tennessee, Chattanooga, TN 37403, USA; 3CQM—Centro de Química da Madeira (Madeira Chemistry Center), Universidade da Madeira (University of Madeira), Campus da Penteada (Penteada Campus), 9020-105 Funchal, Portugal; 4NanoStruc Research Group, Chemistry Department, Faculty of Science, Helwan University, Cairo 11795, Egypt; 5School of Chemical Sciences, Dublin City University, D09 V209 Dublin, Ireland

**Keywords:** Janus particle, drug delivery, formulation, pharmacokinetics, pharmacodynamics

## Abstract

Janus particles have emerged as a novel and smart material that could improve pharmaceutical formulation, drug delivery, and theranostics. Janus particles have two distinct compartments that differ in functionality, physicochemical properties, and morphological characteristics, among other conventional particles. Recently, Janus particles have attracted considerable attention as effective particulate drug delivery systems as they can accommodate two opposing pharmaceutical agents that can be engineered at the molecular level to achieve better target affinity, lower drug dosage to achieve a therapeutic effect, and controlled drug release with improved pharmacokinetics and pharmacodynamics. This article discusses the development of Janus particles for tailored and improved delivery of pharmaceutical agents for diabetes treatment and antimicrobial applications. It provides an account of advances in the synthesis of Janus particles from various materials using different approaches. It appraises Janus particles as a promising particulate system with the potential to improve conventional delivery systems, providing a better loading capacity and targeting specificity whilst promoting multi-drugs loading and single-dose-drug administration.

## 1. Introduction

Janus particles are a special type of nanoparticles or microparticles with surfaces having two or more different physical properties [1]. The term “Janus particle” was coined by Leonard Wibberley in his novel named ‘The Mouse on the Moon’ in 1962 as a science fiction device for space travel [2]. In 1991, Pierre-Gilles de Gennes mentioned the term “Janus” particles in his Nobel lecture, which popularized the term [3]. Janus particles are identified to be advantageous for novel biomedical applications over the past two decades due to their exclusive anisotropic properties, the synergistic ability for combinatorial therapies, and potential multilevel targeting effects [1]. Thus, Janus particles have been developed as an ideal single-carrier system for multiple drugs by dissolving them in solvents of distinct solubilities, which has been attributed to their unique properties of functionalization and asymmetric structure [4].

Janus particles react similarly to amphiphilic molecules with a spatial separation of two physiochemically as well as functionally distinct parts on their surfaces and also within the particles [5]. Thus, a Janus particle can bear hydrophilic groups on one-half of its surface and hydrophobic groups on the other half, resulting in a multicompartmental surface with unique physical and chemical properties [6]. The novel surface of Janus particles makes it possible to incorporate a variety of active ingredients into their structure [7]. Moreover, the simplest form of Janus particles can be developed by dividing the particle into two separate compartments, each composed of distinct materials and/or possessing different functional groups, as well as physical and chemical properties [8]. Consequently, anisotropic or separate compartments can be created on two sides of the particle [9]. In short, Janus particles possess multiple characteristics in terms of chemical composition, shape, polarity, and other physicochemical properties [10]. These unique features of Janus particles enhance their potential applications in various fields, including controlled drug delivery, nanocatalysis, diagnostics, and separation technology [11,12].

In general, Janus particles are classified into polymer-polymer, polymer-inorganic, lipid, metallic, organic-inorganic, organic-organic, and inorganic-inorganic particles based on the materials used for their synthesis [10]. Moreover, Janus particles are synthesized by various techniques, such as solvent evaporation, polymerization, self-assembly, microfluidics, masking, and phase separation [13]. In addition, they can be designed or tuned to have interesting properties, such as magnetic, surface, and optical features [14]. They can also be fabricated to exhibit a wide range of morphologies, including spherical, non-spherical, mushroom, snowman, and dumbbell shapes. Hence, Janus particles are gaining significant attention as promising bio-agents and are fabricated in nano size with uniform distribution [11].

This article discusses novel concepts in the synthesis of various Janus particles from different materials (in both micro and nano sizes), focusing on their varying structural and functional characteristics for tailored and improved pharmaceutical applications. The potential of Janus particles for diabetes treatment and antimicrobial applications is also discussed.

## 2. Overview of Janus Particles

Janus particles were discovered and developed in the search for a novel drug vehicle that can accommodate multiple drugs via double emulsion systems. This class of particles refers to colloidal particles with anisotropic properties and an asymmetric geometry due to chemical and/or polarity variations [15,16]. Figure 1 shows the variety of structural conformations in anisotropic Janus particles. Cho and Lee (1985) reported and generated the first Janus nanoparticles using poly(methyl methacrylate)-polystyrene composite particles via a seeded emulsion polymerization approach [17]. Subsequently, Casagrande and his team developed spherical glass particles called “Janus beads”, which consist of hydrophobic and hydrophilic hemispheres and exhibit specific properties at their water-oil interfaces, compared with other solid particles [18].

In general, Janus particles are divided into two main groups, such as compartmentalized and patchy forms, based on their morphology. Compartmentalized Janus is a complex particle consisting of various phase-divided domains within a core, whereas patchy Janus particles have precisely controlled patches with different surface structures [20]. The advantageous properties and functions of Janus particles in applications have not been completely exploited due to the complications in the successful synthesis of particles of controlled size, purity, and shape. In addition, the interparticle interaction at fluid interfaces as shown in Figure 2 has been identified to be essential for the stability of Janus particles. The advancements in material synthesis and characterization research have led to numerous techniques to produce Janus particles, including solvent evaporation, polymerization, self-assembly, microfluidics, masking, and phase separation. Table 1 provides an overview of the different methods used to prepare Janus particles for various applications.

### Structural and Functional Characteristics of Janus Particles

Janus particles are structurally distinguished by the presence of two distinct hemispheres with unique properties, such as morphologies, physiochemical characteristics, and functionalities in a single entity [29]. Numerous Janus particles are recently produced with controllable chemical and topological anisotropy. For example, Janus particles with one hemisphere of cationic surface charge and the other with anionic surface charges were synthesized as displayed in Figure 3. This is possible by precise adjustment of the nature and concentration of the monomer under optimized polymerization conditions, resulting in an enriched Janus community with different charges (positive, negative, or neutral). In addition, various fabrication techniques enable the formation of Janus particles with exclusive sizes ranging from the nanoscale to the microscale [30]. For instance, crosslinking and self-assembly methods enable the formation of nano-sized Janus NP, while microfluidic techniques can produce micron-sized Janus formulations [31]. Various fabrication techniques and their significance are briefly discussed in Section 3. Janus particles can be fabricated in a variety of shapes, such as rods, tubes, disks, cylinders, spheres, ellipses, acorn-like shapes, and hamburger-like shapes [31,32]. Their bifunctional properties offer significant opportunities for broad applications in the pharmaceutical and biomedical industries to improve disease diagnosis, increase the efficiency of drug encapsulation and target specific cellular sites [33]. Similarly, Janus particles can also be used in applications, such as textiles, sensors, bioimaging, and catalysis [14]. 

It is worth noting that different structures of Janus particles can lead to unique physicochemical and functional properties [35]. For example, an inorganic-polymer hybrid can be used to improve the surface functionalization, biocompatibility, and stability of Janus particles [36,37]. Further, Janus particles composed of purely inorganic materials, such as inorganic-inorganic complexes, are promising candidates for biomedical applications as their region-specific functionalization makes them an ideal tool for bioimaging, in vivo diagnostics, delivery, and treatment [38,39,40]. Polymer-polymer hybrids are highly desirable as Janus particles, as they can be easily tuned to exhibit a variety of shapes and surface properties [41]. Winkler et al. (2019) synthesized bicompartmental poly(lactic-*co*-glycol)-(PLGA)/polycaprolactone (PCL) Janus particles for the successful delivery of both hydrophilic and hydrophobic drug molecules. They applied a novel double emulsion technique to encapsulate two drugs with different water solubility into biocompatible Janus particles and deliver them together. They demonstrated staggered release of the drug molecules with independent release kinetics. The staggered release of drugs is particularly advantageous in treatments that require the administration of multiple drugs sequentially at specific rates over a limited period. This allows pre-programmed release characteristics of Janus particles to meet different therapeutic requirements. Results showed excellent encapsulation efficiency (85–94% for hydrophobic and 68% for hydrophilic agents), successful compartmentalization of agents within the carrier to avoid interference, and controlled release of two agents over time [15] as shown in Figure 4.

In another study, advances in dual-reacting Janus particles for biomedical applications have been demonstrated by the carriage and release of payloads with stimuli-only behavior. This release pattern cannot be achieved with core-shell or monomorphic nanoparticles. A thermo-reactive polymer named poly(2-hydroxyethyl methacrylate) (PHEMA) and a pH-sensitive polyelectrolyte called poly(2-dimethylaminoethyl methacrylate) (PDAMEMA) were used as carriers for doxorubicin and ibuprofen, respectively. The resultant Janus particles have two different morphologies (snowman-like and dumbbell-like shapes), with different loading capacities for both drugs. The study showed that doxorubicin was rapidly released under low pH conditions due to its high solubility, while the release of ibuprofen was higher under neutral pH conditions (~7.4) [43]. This tunable property is particularly useful for drug delivery and release at specific cellular targets, such as the endoplasmic reticulum and intestine. Thus, it is evident that the drug release by Janus particles is more advanced, compared to other conventional drug release systems, and is beneficial for desired therapeutic applications [44,45,46,47].

Other studies have also shown the promising potential of Janus particles. Studies show gold-nickel Janus nanorods for non-viral delivery of vaccines and genes with higher transfection efficiency [48]; polymer-inorganic Janus particles composed of gold nanospheres and poly(styrene)-block poly(acrylic acid) (PS-PAA) to target tumor cells [49]; and multifunctional Janus nanoparticles based on superparamagnetic iron oxide nanoparticles (SPION) for the treatment of glioblastoma which suggested that Janus nanoparticles are a potential candidate for the development of new therapeutic formulations and delivery methods for the treatment of brain cancer [50]. Further, Janus nanoparticles of gold-iron oxide are used as imaging agents for in vivo multimodal imaging due to their anisotropic structure [51]; and Janus nanoparticles of gold-mesoporous silica loaded with doxorubicin are used for theranostics and selective chemotherapy in addition to specific imaging and drug release monitoring application [52].

## 3. Fabrication of Janus Particles

Masking, self-assembly, microfluidic approach, and phase separation are the common methods for the fabrication of Janus particles, as summarized in Figure 5.

### 3.1. Masking

Masking is a manufacturing technique based on the chemical alteration of one side or compartment of particles to create asymmetric Janus particles. It is a simple method that allows the coupling of different functional groups to generate Janus particles. In this method, Janus particles are formed when the uncovered hemisphere is masked by chemical reagents, which later trigger their unique properties, or when the particles are confined at the interface of two immiscible mixtures [54]. The masking procedure for the formation of Janus particles consists of four main steps, as shown in Figure 6, such as (i) exposure of a hemisphere of homogeneous nanoparticles; (ii) application of masking techniques, such as evaporative deposition and suspension of nanoparticles at the interface of two phases, where the chemically inert “blocking” surfaces can be either solid (e.g., polymer crystal) or liquid (e.g., Pickering emulsion) [8,55]; (iii) chemical modification of particle properties by the masking process; and (iv) removal of the masking agent, resulting in the formation of Janus particles [5].

### 3.2. Self-Assembly

The self-assembly approach is widely used for the competitive surface assimilation of incompatible ligands, such as hydrophobic and hydrophilic ligands, on particle surfaces and for block copolymer formation, including di- and tri-block copolymers as summarized in Figure 7 [56,57,58]. Gold nanocrystals with their thiol-binding affinity are widely used to perform competitive adsorption of incompatible ligands onto particles via self-assembly [59]. However, there have been few reports on this type of self-assembly technique. Self-assembly of block copolymers usually occurs by radical polymerization. The process is initiated by preparing the copolymers in an ordinary solvent and modifying the solvent to proceed with the self-assembly process [11,60]. The effectiveness of the self-assembly process can be influenced by several factors, including pH, ionic strength, and temperature [61]. Generally, self-assembly is limited to a few sample quantities, as it is unstable at high concentrations of copolymers. The first tri-block copolymers used to form Janus particles were presented by Erhardt and his team in 2001. They developed Janus particles with a southern and a northern hemisphere along with a cross-linked core using techniques, such as solution casting, cross-linking, and redissolution processes in a selected solvent [62]. The solution casting step has led to the preparation of ABC-type triblock copolymers with embedded spherical domains of the central block, crosslinking the spherical domains and using a good solvent to redissolve the main phase. The results demonstrated an equilibrium between the molecularly dissolved Janus micelles and aggregates. Moreover, this study showed that Janus particles can be generated by preserving the molecular superstructure of a microphase-separated ABC triblock copolymer.

### 3.3. Microfluidic

Microfluidics is a method that requires an organic solvent (dispersed phase) to dissolve both hydrophilic and hydrophobic compounds. Later, the dispersed phase is injected into an aqueous phase to create amphiphilic droplets by controlling the flow rate of both the dispersed and aqueous phases. This technique allows equal quantities of different polymers to be encapsulated in the resultant droplet as the dispersing phase exerts the same shear force, resulting in the formation of Janus beads. Subsequently, polymerization is initiated by thermal initiation or UV irradiation [11]. A modified fluidic nanoprecipitation system was used to generate PLGA Janus particles with both hydrophobic and hydrophilic agents in exclusive compartments. The system allowed for a one-step manufacturing approach with two inlets for the Janus components to be introduced into the precipitation stream, as shown in Figure 8. In this method, the release properties of hydrophobic agents were improved, compared with monomorphic particles [64]. In addition, a simple droplet-based microfluidic fabrication technique was developed to produce hybrid (gold nanorods @ silver)-polyaniline (PANI) Janus nanoparticles, which was identified to be beneficial as sensors for surface-enhanced Raman scattering application. The technique enabled the production of Janus particles with uniform size, excellent dispersion, and short response time [65].

### 3.4. Phase Separation

The concept of the phase separation technique is based on an oil-in-water emulsion, where the oil phase consists of two incompatible polymers, that are homogeneously mixed with the help of a co-solvent and dispersed dropwise in a water phase, followed by precipitation of Janus particles after evaporation of the solvent as summarized in Figure 9 [5,66]. This approach can produce Janus particles with excellent colloidal stability. Liu and team (2013) reported the preparation of non-spherical and amphiphilic Janus particles by a reaction-controlled phase separation technique with kinetic control [67]. In this study, a direct synthesis technique to prepare hydrophilic inorganic silica/hydrophobic polystyrene (PS) polymeric Janus particles using a mixture of styrene (St) and octadecyltrimethoxylsilane (ODTS) was developed. The structural features of the particles could be tuned by modification and kinetic control of ODTS hydrolysis and St polymerization. This technique is scalable and applicable to all mixtures, including organic/organic, inorganic/inorganic, and organic/inorganic systems [5]. Emulsion polymerization is a partial phase separation technique that enables colloidal dispersions of latex polymer formulations in water with improved stability. In addition, emulsion polymerization allows better control of the morphological properties of the particles, and the particle size is usually in the range of 0.05 to 1 µm [68,69].

### 3.5. Factors Affecting the Fabrication of Janus Particles

Several factors must be considered while synthesizing Janus particles with symmetrical compartments, which include the degree of incompatibility and the molecular weights of the two compounds, as well as the interfacial tension at the water-solvent interface. It has been reported that compounds with a high degree of incompatibility and a high molecular weight in addition to a low interfacial tension can produce more symmetric Janus particles [71,72]. Thus, the efficiency of the self-assembly process can be influenced by various factors, such as pH, ionic strength, and temperature. The “Janus equilibrium” is also a significant factor to be considered in the synthesis of perfectly biphasic Janus particles with the desired anisotropic structures [73]. Various fabrication techniques can be used to control the physical properties, surface chemistry, and material composition to produce the desired Janus particles [74]. Nucleation growth and surface nucleation methods are primarily used to generate complex Janus morphologies [75]. Other factors can influence the effectiveness of the various manufacturing processes. For example, the conductivity of polymer masks plays a critical role in achieving evenly distributed compartments on the Janus microspheres [7]. The slow feed rate of the monomers and the high degree of crosslinking are also essential parameters that affect the structure of the resultant anisotropic particles [76].

## 4. Biomedical Applications of Janus Particles

Recently, several effective drugs have become available on the pharmaceutical market. However, their efficacy in the treatment of common diseases, such as cancer, diabetes, and neurodegenerative diseases is still hampered by factors, including low targeting, systemic cytotoxicity to other normal cells, poor solubility and hydrophobicity, excessive and frequent dosing, and rapid renal filtration [77,78,79]. Therefore, it is crucial to design the drug delivery system at the molecular level to improve its therapeutic indices, targeting, pharmacokinetics, and pharmacodynamics. Hence, Janus particles have been introduced as an effective system for targeted and controlled drug delivery. In addition, they are also used as biomarkers for targeted cancer chemotherapy and various other biomedical applications. Table 2 provides an overview of various Janus particles that have been extensively studied for other biomedical applications.

## 5. Janus Particles as a Potential Delivery System: A Case Study in Diabetes Theranostics

Diabetes is a major health problem, and according to the World Health Organization (WHO), more than 366 million cases of diabetes complications are expected worldwide by 2030 [87]. Generally, diabetes is classified into type 1 diabetes (insulin-dependent diabetes), type 2 diabetes (insulin-independent diabetes), and gestational diabetes, with 90% of diabetes cases belonging to type 2 [88,89]. There are several treatment methods, such as insulin therapy and islet transplantation, to cure diabetes [90,91,92,93,94], although some of these methods have significant challenges. Currently, most of the available diabetes therapies are based on adequate glycemic as well as lipid control and reduction of complications [95]. The advancements in biomedical technology have shifted the research focus to the use of smart drug delivery systems and the promising potential of Janus particles in the development of drug formulations and delivery strategies for the effective treatment of diabetes [96].

Among the various applications of Janus particles for drug delivery, it has been utilized for the treatment of diabetes [97]. In this section, the use of Janus particles for the treatment of diabetes as an efficient drug delivery system is described as a case study. For example, a Janus microparticle drug delivery system has been identified that provides sustained release of rapamycin (an immunosuppressant) for 30 days through the anterior chamber of the eye to suppress immune rejection of transplanted pancreatic islets in diabetic patients. These Janus microparticles were prepared from biopolymers, such as poly(lactic-*co*-glycolic acid) (PLGA) and polycaprolactone (PCL) by a one-step emulsion-solvent evaporation technique [98,99]. In another study, the preparation of polymeric Janus particles using solvent emulsion was also demonstrated to overcome the challenge of low throughput in microfluidics [100]. In this work, the encapsulation of rapamycin and glibenclamide with two drugs was demonstrated to protect pancreatic islet grafts and trigger insulin release from islets. The Janus particle system was able to release two drugs simultaneously to overcome the challenges of islet transplantation.

Another Janus particle, composed of the biopolymer ethylene vinyl acetate (EVA)/poly(L-lactide) (PLLA), was used to encapsulate superparamagnetic iron oxide nanoparticles (SPIONs) in addition to rapamycin and glibenclamide. The use of SPIONs to release drugs selectively represents a novel and potentially effective method for diabetes treatment. The encapsulated SPIONs triggered the release of glibenclamide from Janus particles via the application of radiofrequency, especially during meal ingestion, and suppressed insulin secretion, when it is not required [100,101]. This work also demonstrated the great potential of Janus particles to improve glycemic control with less insulin dependence.

### 5.1. Biosensor for Diabetes Diagnosis

Janus particles have been extensively studied to improve the targeted diagnosis of diabetes, as indicated in Figure 10. Diez and Sanchez (2014) [102] reported an intelligent delivery system consisting of Janus nanoparticles with opposing surfaces of mesoporous silica (MS) and gold (Au), which act like a Boolean logic OR gate. The MS surface was coupled with a pH-stimulating β-cyclodextrin-based supramolecular nano-valve, while the gold surface acted as a “control unit” and was functionalized with esterase and glucose oxidase enzymes as effectors. The release of the encapsulated drug molecules was controlled by this pH-sensitive Janus nano-valve with an enzymatic operator, which was sensitive to the specific enzyme substrates D-glucose and ethyl butyrate. The nano-valve on the surface MS of the Janus nanoparticles was opened for the release of the drug only after it was triggered by a lower pH resulting from the oxidation of esterase and glucose oxidase into their acidic forms [102]. In another study by Lu et al. (2015) [103], the multifunctionality of Janus hematite-silica-γ-Fe_2_O_3_/SiO_2_ nanoparticles (JFSNs) as a biosensor for the detection of glucose and hydrogen peroxidase was demonstrated by colorimetric measurement over a wider range of pH and temperature with great stability. The study demonstrated that JFSNs with asymmetric properties can be immobilized with glucose oxidase to provide highly selective and reproducible colorimetric detection of glucose, exhibiting the catalytic effect of both glucose oxidase and peroxidase. Consequently, Janus particles can be used to integrate two different detection systems as a multifunctional glucose biosensor to diagnose glucose content in samples such as blood serum [103]. In addition, fluorescent Janus microspheres have been demonstrated for glucose detection and pH calibration, which are suitable as implantable sensors for continuous monitoring of glucose [104]. The fluorescent Janus microspheres were prepared using a UV-assisted centrifugal microfluidic instrument and contain a pH sensor and a fluorescent glucose sensor in each hemisphere as a multimodal monitoring device for diabetes management.

### 5.2. Delivery System for Anti-Diabetic Drugs

There are a variety of antidiabetic drugs that have been developed to treat diabetes through different mechanisms, either to increase insulin levels in the body or to achieve successful islet transplantation. Rapamycin is a hydrophobic immunosuppressant that is widely used in chemotherapeutic treatments and the suppression of post-transplant rejection. Rapamycin acts by moderating the body’s immune system to suppress immune rejection of the graft for post-transplant diabetes treatment [99,106]. Dexamethasone is a corticosteroid hormone, called a glucocorticoid, which is hydrophobic and generally used in ophthalmology as an anti-inflammatory drug. It is slightly lipophilic and has a low molecular weight, which increases membrane penetration and permeability. It acts as an immunosuppressant by lowering the natural defenses of the immune system to minimize undesirable conditions, such as allergic reactions and swelling behavior. It is a promising candidate for cell therapy and transplantation to prevent immune rejection of islet grafts in the treatment of diabetes. The therapeutic effect of dexamethasone as an immunosuppressant can be observed within one day and lasts for about 3 days [107,108].

Further, Janus particles can facilitate the long-term release of immunosuppressants such as rapamycin and dexamethasone when co-administered with islet transplantation. They allow sustained release of immunosuppressants for up to 30 days, which is twice the duration of natural islet cell rejection. This is achieved by controlling the mixing ratio of the particles, the interfacial tension, and the spreading coefficient. Janus particles represent a smart engineering approach to increase the solubility limit, slow the degradation of immunosuppressants, and maintain the minimum therapeutic dose via the right biopolymers with high loading capacity and slow degradation rate. For example, the half-life of rapamycin in the blood is 15 h. It is possible to release the drug continuously in the blood for three weeks with biopolymers, such as PLGA [99,109,110]. Gilbenclamide is an antidiabetic drug used to treat type 2 diabetes by oral administration. It is a diaryl sulfonyl urea that has high oral absorption but low dissolution in gastric fluid. The pharmacokinetics of glibenclamide is regulated by solute carriers (SLC), organic anion-transporting polypeptides, and transporters of the ABC superfamily. Its metabolism is modulated in the liver by cytochrome p450-regulated oxidative metabolism, while subsequent distribution and excretion are regulated by adenosine triphosphate (ATP)-binding cassette transporters (ABC) [111]. In addition, the neurotransmitter beta-aminobutyric acid (GABA) has been reported to have the potential to trigger the formation of new beta cells and maintain the population of effective beta cells. Studies have demonstrated the significance of GABA in both type 1 and type 2 diabetes, where GABA can bind and interact with γ-aminobutyric acid A (GABAA) receptors on specific ion channels, leading to effective regulation of insulin secretion [112]. Further, GABA has anti-inflammatory effects in pancreatic islets to increase beta cell survival and decrease the number of toxic white blood cells [113]. Moreover, glucagon-like peptide (GLP−1) is an incretin hormone that is lower in diabetic patients. GLP−1 binds to GLP−1 receptors, that normally exist in tissues, such as pancreatic ducts, pancreatic beta cells, skin, immune cells, lung, kidney, and gastric mucosa. Therefore, GLP−1 agonists can be used as a drug in diabetes treatment by mimicking the hormone GLP−1 and binding to GLP−1 receptors on beta cells, which stimulates glucose-dependent insulin secretion from pancreatic islets, leading to a reduction in blood glucose levels. The GLP−1 agonist acts by regulating various ion channels such as voltage-gated K^+^ and Ca^2+^ channels, ATP-sensitive K^+^ channels, and non-selective cation channels. It also regulates exocytosis and intracellular energy homeostasis and stimulates islet cell growth, insulin gene transcription, and neogenesis [114]. It has been reported that the synergistic effects of GABA and GLP−1 may contribute to the prevention of diabetes by preserving pancreatic beta cell mass and reversing apoptosis, compared with the use of a standalone drug [115].

In a recent study by Fan et al. (2019) [106], novel rapamycin-loaded microparticles were developed to promote the survival of co-injected islet-allografts in the anterior chamber of the eye (ACE) for the treatment of type 2 diabetes. ACE in the anterior chamber of the eye is reportedly a better site for islet transplantation, compared to the conventional intrahepatic site, as it is more accessible, easy to monitor, minimizes surgical invasiveness, allows local administration of immunosuppressants, such as rapamycin, and has a high survival rate after transplantation. The results demonstrated the efficacy of the developed microparticles using United States Food and Drug Administration (USFDA)-approved biopolymers, such as PLGA and PCL in the controlled and sustained release of rapamycin to achieve local immunosuppression against islet cell rejection during islet graft re-vascularization [106]. The study showed that the rapamycin-loaded microparticles preserved allogeneic islet grafts for up to 30 days and improved insulin dependence and glycemic control.

Innovative Janus particles with distinct and independent compartments are an intelligent delivery system that can release both γ-aminobutyric acid (GABA) and glucagon-like peptide 1 (GLP-1) either simultaneously or sequentially at desired time intervals, allowing spatially controlled encapsulation of different functionalities. This can be achieved by the proper selection of polymers to form asymmetric structures that can control the loading capacity and release pattern of different bioactive ingredients with incompatible properties in a single particle [11]. Recently, Bao et al. (2020) [116] reported the preparation of a Janus membrane modified with micropores as a self-pumping bioactive wound dressing composite for effective diabetes wound healing applications. The resultant Janus membrane demonstrated an improved ability to allow the transport of wound exudate from the wound surface to the dressing material and allow controlled reflux of bioactive ions containing wound surface fluid to simulate angiogenesis [116], as shown in Figure 11. In addition, Janus particles allow real-time monitoring of sequential drug release from different compartments, enabling independent release kinetics that are unattainable with single nanoparticles. This promotes multiple and synergistic effects of combined therapies and the ability for multiple targeting, which is impossible in isotropic systems [117]. It is noteworthy that the ability of Janus particles to have surface hemispheres with two different properties allows them to have a binding link for the surface delivery of the drug [12,118]. Moreover, reducing the size of Janus nanoparticles plays a significant role in improving their efficacy in drug delivery [38,119,120].

## 6. Janus Particles as a Potential Delivery System: A Case Study in Antimicrobial Theranostics

Recently, several Janus particles were identified to have antimicrobial activity and were either incorporated into textiles or used as microbial inhibitors. Hou et al. (2021) [121] synthesized novel Janus particles containing poly(vinylidene fluoride) titanium dioxide nanoparticles on one side and epoxy resin on the other side using the electrospray method and coated them onto the surface of the fabric. The study revealed that the irregular hierarchically coated Janus particles on the tissue surface can inhibit 74.8% of *E. coli* in 24 h, along with UV protection properties (ultraviolet protection factor (UPF)—733) and high hydrophobicity with a contact angle of 152°, which can be attributed to the shape of Janus particles [121]. In addition, Panwar et al. (2018) [122] prepared novel silver-silica Janus particles with different functional groups such as thiol, amine, and epoxide on the surface of silica particles by the Pickering emulsion method, as shown in Figure 12. The resulting particles had a size of ~3 nm, provided a surface for silver nanoparticle attachment, and were resistant to agglomeration with enhanced antimicrobial activity against *Staphylococcus aureus* after being introduced into a cotton fabric [122]. Moreover, Lozhkomoev et al. (2018) [123] demonstrated the synthesis of iron-copper and iron-silver bimetallic nanoparticles by electrical explosion of wires with a Janus structure and a boundary between the two metal phases. The results showed that the bimetallic nanoparticles have enhanced antibacterial activity against Gram-positive *Staphylococcus aureus* and Gram-negative *Pseudomonas aeruginosa* due to their Janus structure. In addition, the Janus structure altered the dissolution rate of iron in a sodium-phosphate buffer solution, with a higher dissolution rate for iron-copper samples than for iron-silver nanoparticle samples [123].

Jia et al. (2015) [124] synthesized Janus particles of silver and chitosan, which exhibited potential antimicrobial activity. The study identified that the Janus particles with a concentration of 0.02 mg/mL had an enhanced ability to suppress the growth of *Botrytis cinerea* fungi. In addition, these materials also showed antibacterial activity against *Bacillus subtilis*, *E. coli*, *Staphylococcus aureus*, and *Salmonella choleraesuis* [124]. Additionally, Chen et al. (2020) [125] used redox and click chemistry for the design and synthesis of mesoporous silica-silver Janus nanoparticles modified with cardanol. The study showed that the Janus particles have 99% antibacterial activity against *S. aureus* and *E. coli* due to the release of silver ions and the phenolic hydroxyl groups of cardanol. Moreover, these structures were found to have little or no cytotoxicity [125]. All these studies showed that the antimicrobial property is either present or enhanced due to the anisotropic Janus structure compared with other holistic structures.

Apart from antimicrobial efficacy, Janus particles possess enhanced biosensing ability. Recently, Das et al. (2021) used Janus particles prepared from fluorescent polystyrene particles (1 micron) for the detection of *E. coli* pathogens by rotational diffusometry through the amplification of specific genes. The detection was found to be specific for *E. coli* DNA at 50 pg/µL. This technique was shown to be superior to a conventional polymerase chain reaction (PCR) with an improved detection time of less than 60 min [126]. Further, Sinn et al. (2012) prepared a novel 300 nm nickel, spin coated with 10 µm of polystyrene particles onto a 4” glass wafer, magnetized with 60 mT of permanent magnetic field for 3 days and suspended in phosphate-buffered saline for the formation of magnetically uniform Janus particles. The resultant Janus particles were utilized for the preparation of asynchronous label-free magnetic bead rotation (AMBR)-based viscometry to detect bacterial growth. The study showed that the novel viscometry method is effective in detecting the growth of uropathogenic *E. coli* isolates of initial 50 cells per drop of concentration within 20 min [127]. Furthermore, Chun et al. (2018) demonstrated the benefits of retroreflective Janus particles as a potential non-spectroscopic signaling probe for loop-mediated isothermal amplification (LAMP)-based detection of *Salmonella typhimurium* as shown in Figure 13. In this study, the Janus particles were prepared by a sequential coating of gold (20 nm) and aluminum (40 nm) on the surface of hemispherical silica microparticles (1.2 µm) via physical vapor deposition. The study showed that this novel strategy is better than conventional microbial sensing methods, such as polymerase chain reaction (PCR), real-time PCR, and LAMP for quantitative detection of *S. typhimurium* with a limit of detection (LOD) of 10^2^ colony-forming unit (CFU) [128]. Thus, Janus materials can also be used as a potential biosensor for pathogen detection.

## 7. Prospects and Challenges of Janus Particles as Drug Delivery Vehicle

### 7.1. Prospects

Although Janus particles have gained attention only in recent times, they have been used for pharmaceutical applications and have been under extensive research. For example, certain commercially available bilayer drug tablets are manufactured in the Janus dosage form to encapsulate and deliver two different or incompatible active pharmaceutical ingredients (APIs) for different therapeutic effects [129,130]. Janus particles are used as a new generation of anisotropic nanoparticles with promising properties for drug delivery with biphasic release kinetics, different target affinity, and opposite solubility/charge [83]. Conventional drug carriers are only capable of co-administering two drugs with similar properties, for example, two hydrophilic or hydrophobic drugs. Janus particles are more suitable for the co-administration of drugs with different chemical properties due to their anisotropic properties and dual functionality and/or physicochemical property [131]. One of the significant features of Janus particles is their ability to allow the complexation of two different drugs in distinct compartments within a system, resulting in simultaneous loading, transport, and release of various drug molecules at the cell target site with controlled release and good pharmacokinetics [4,15].

Moreover, Janus particles offer tremendous superiority in terms of size, surface properties, and shape, and provide the desired biodistribution, pharmacodynamics, and pharmacokinetics [132]. Their large surface-area-to-volume ratio improves targeting accuracy and binding ability and maximizes the number of target elements and the area of protective coating material [133]. Both contrast agents and drugs can be loaded separately to allow real-time diagnosis and tracking of therapeutic events due to their anisotropic nature [134]. In addition, Janus particles with complex geometries are resistant to rapid filtration through the reticuloendothelial system [135]. Moreover, Djohari and Dormidontova (2009) [136] showed that Janus nanoparticles can provide higher targeting accuracy and selectivity compared with conventional nanoparticles with homogeneously distributed targeting ligands. The study showed that anisotropic or isotropic Janus nanoparticles have a center of mass closer to the receptors on the surface and have a more compact shape. They are also more stable, being able to reach an equilibrium amount of bound targeting ligands in a shorter time, compared to conventional nanoparticles with a similar quantity of ligands [136]. The results emphasized that Janus nanoparticles exhibit faster binding interaction with cell surface target receptors. They are an ideal candidate for theranostic applications as they can encapsulate both diagnostic and therapeutic agents in the same particle.

Janus particles represent a unique innovation in the field of biomedical materials and enable the application of biodegradable Janus particles for in vivo bioimaging and therapeutic delivery, as they can be engineered to have minimal/no toxicity [137]. For example, the concentration of biomaterials used to develop Janus particles must be carefully controlled to prevent excessive bioaccumulation [138]. Additionally, hollow and/or porous Janus particles provide a high drug-loading capacity, allowing them to act as both drug carriers and contractile agents [139]. This provides a promising avenue for developing real-time monitoring technologies and improving therapeutic efficacy. However, further research efforts are needed to better understand the functioning of Janus particles and their interactions with surrounding biological systems via their structural anisotropy to promote their biomedical applications in the future.

### 7.2. Challenges

Challenges still associated with Janus particles include nanoscale size control, development of Janus particles with uniform shape, scalability of production at high yield, low production cost, and regulatory hurdles [5]. It is significant to overcome the challenges of Janus particles, including issues related to the interfacial and bulk properties of the manufactured particles, which are not generally recognized as safe (GRAS) and are not approved for use in pharmacopeia to enable widespread pharmaceutical application [14]. This is often associated with the use of dendrimers or di-/tri-block copolymers [140]. “Janus equilibrium” is another factor that must be considered in the characterization of Janus particles and remains a major challenge in the preparation of perfect biphasic Janus particles with the desired anisotropic shape [141]. Hence, research interest has shifted to the development of asymmetric Janus particles, such as colloidal structures with multiple patches or compartments that do not exhibit complementary biphasic conformations [142]. The geometry of Janus particles is also a crucial parameter that needs to be controlled during the synthesis process to produce the effective ratios of each compartment for different applications [143]. Another challenge is the effective morphological characterization of Janus particles. The characterization requires more advanced imaging technologies to study the nanoscale and anisotropic Janus particles. For example, the combination of different tools such as small-angle X-ray radiation, resonance energy transfer, neutron scattering, and cryo-transmission electron microscopy is used to image nanometric Janus particles with two compartments, which exhibits sufficient electron density contrast between the two different compartments to obtain better images [29,144]. Furthermore, it is challenging to achieve thermodynamic equilibrium at the interface between two different compartments when exogenous components such as drugs or biological agents are introduced into the complex [145]. This is mainly due to the interactions between the foreign molecules and the materials of the Janus compartments, which lower the interfacial tension and thus affect the formation of the Janus system [146]. It is also rather rare to obtain nanosized Janus particles (<100 nm), as most of the reports fall in the range of sub-micrometers to a few micrometers [147]. In addition, most manufacturing techniques are lengthy due to the complex nature of the final Janus formulation [148]. There are still certain challenges to be overcome in producing biodegradable and low-toxicity Janus particles with high loading capacity in large quantities [97]. Hence, significant research progress is required to enable the widespread application of Janus particles in various fields.

## 8. Conclusions

Janus particles are a promising platform for the development of tuneable and robust multimeric formulations with different payload capacities and targeting capabilities for different disease conditions. They offer the possibility of combining different pharmaceutical agents in a multimodal treatment to improve therapeutic efficacy. This article focuses on the unique properties and importance of Janus particles as advanced formulations for pharmaceutical applications. The use of Janus particles promises to overcome the drawbacks of conventional delivery systems, such as high therapeutic doses, frequent drug administration, low specificity to target cells, and single drug loading applications. In addition, Janus particles possess several advantages to improve drug efficacy in terms of better solubilization, bioavailability, and pharmacokinetics. Nevertheless, further research efforts are essential to overcome the limitations of Janus particles and enhance their potential for broad theranostic application.

## Figures and Tables

**Figure 1 pharmaceutics-15-00423-f001:**
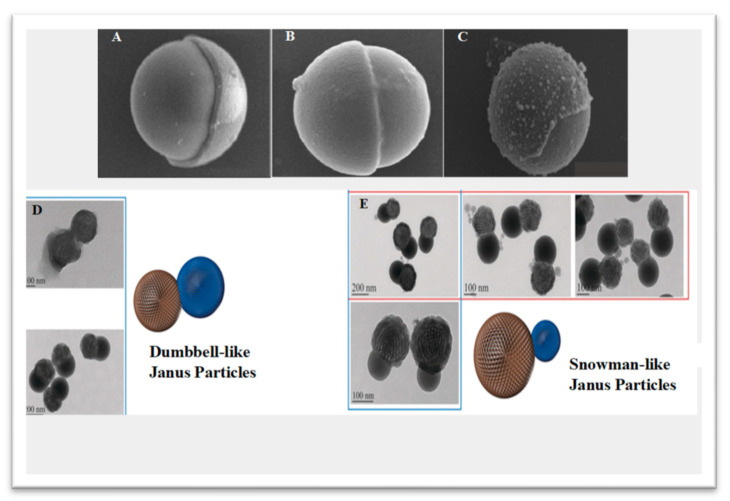
Anisotropic Janus particles with a variety of structural conformations. (**A**–**C**) SEM images of various spherical Janus particles and (**D**,**E**) TEM images of dumbbell-like and snowman-like Janus particles. Reproduced with permission from [19], © Wiley (2017) and © Royal Society of Chemistry (2010).

**Figure 2 pharmaceutics-15-00423-f002:**
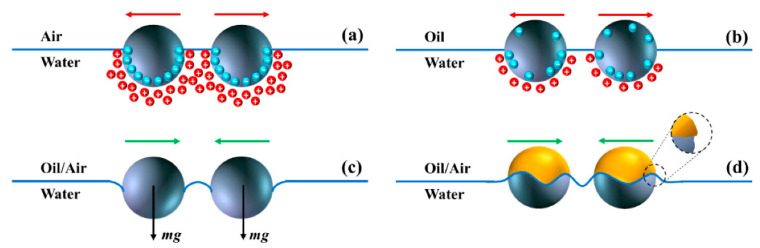
Interparticle interaction of stable Janus particles in fluid interfaces. (**a**) Surface group dissociation leading to dipole-dipole-based repulsive interactions; (**b**) Oil phase-based contact of particle surface for water entrapment leading to long-range repulsive interactions; (**c**) Generate attractive capillary interactions via gravitational forces deformed interface and (**d**) Surface roughness improved by interface undulations to enhance the stability of Janus particles via capillary interaction. Reproduced with permission from Correia et al. (2021), © MDPI, 2021 [21] (Open access).

**Figure 3 pharmaceutics-15-00423-f003:**
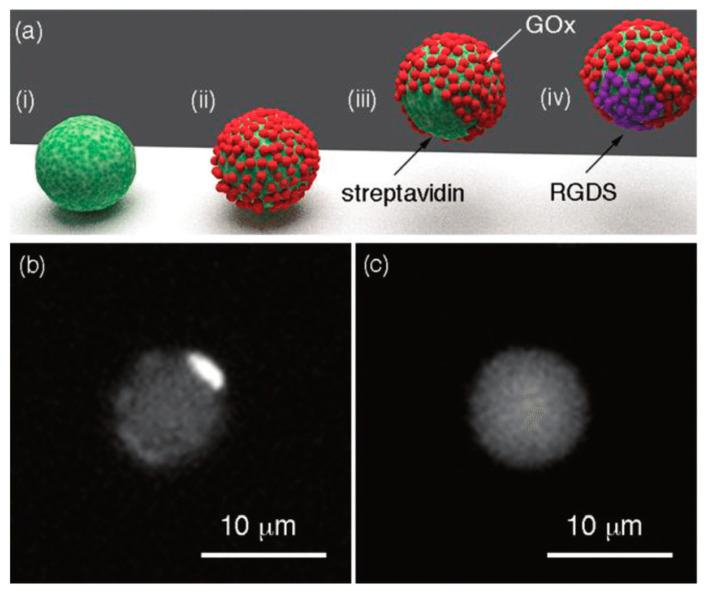
Synthesis and characterization of a specific Janus particle. (**a**) Schematics of glucose oxidase (GOx)-coated Janus particle formation, where (i) streptavidin-coated colloids with a magnet, (ii) Modification of the surface exposed to the solvent with biotinylated enzymes, (iii) Janus particles partially covered with enzymes after removal of the magnet, (iv) Biotinylated arginine-glycine-aspartic acid (RGDS) peptides; Fluorescence microscope image of (**b**) bovine serum albumin and asymmetric arrangement of GOx on the Janus particle surface; and (**c**) Janus particle modified only with GOx. Reproduced with permission from Rucinskaite et al. (2017), © Royal Society of Chemistry (RSC), 2017 [34].

**Figure 4 pharmaceutics-15-00423-f004:**
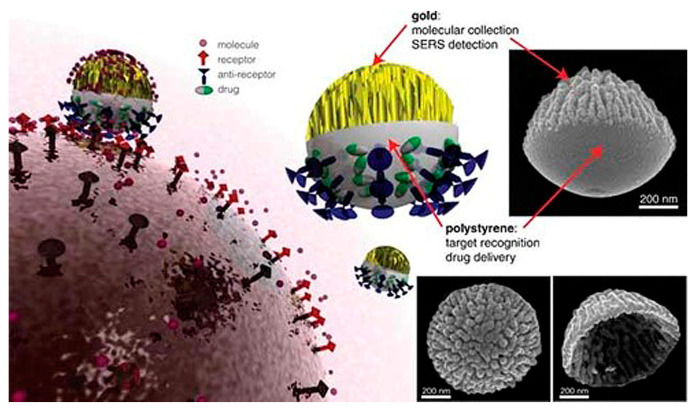
Nanocorals in Janus structure as multifunctional targeting, sensing, and drug delivery nanoprobe; Inset: Scanning electron microscope (SEM) images of Janus structured nano-coral probes. Reproduced with permission from Wu et al. (2010), © Wiley, 2010 [42].

**Figure 5 pharmaceutics-15-00423-f005:**
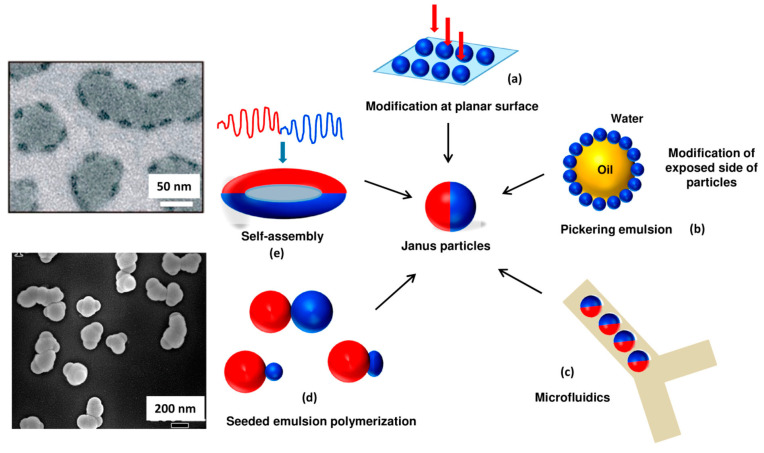
Overview of production approaches of Janus particles. (**a**) Modification and fixation on solid substrates, (**b**) Pickering emulsion method, (**c**) Microfluidic method, (**d**) Seeded emulsion polymerization approach, (**e**) Self-assembly approach. Reproduced with permission from Agarwal and Agarwal (2019), © American Chemical Society, 2019 [53].

**Figure 6 pharmaceutics-15-00423-f006:**
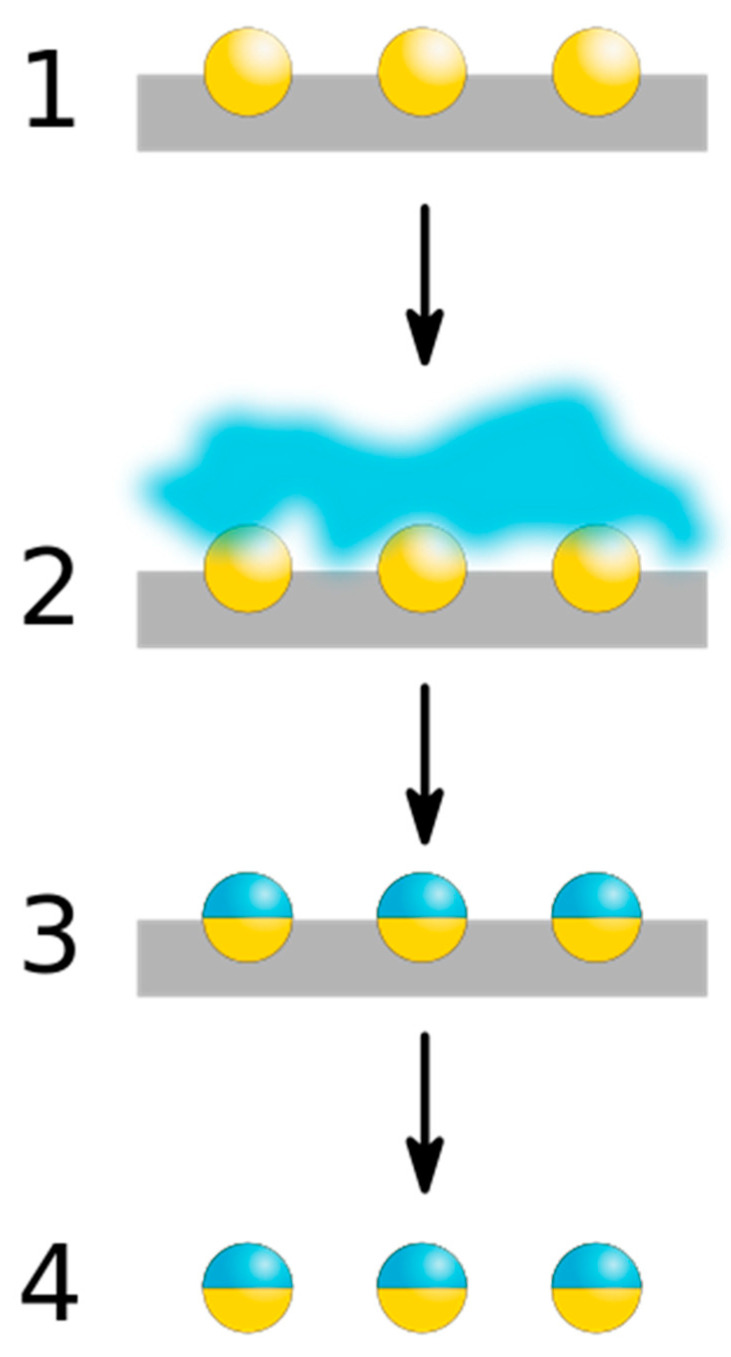
Schematic representation of the crucial steps in the masking technique to form Janus particles. (1) Exposure of a hemisphere of homogeneous nanoparticles; (2) Application of masking techniques; (3) Chemical modification of particle properties via masking process; and (4) Removal of the masking agent, resulting in Janus particle formation.

**Figure 7 pharmaceutics-15-00423-f007:**
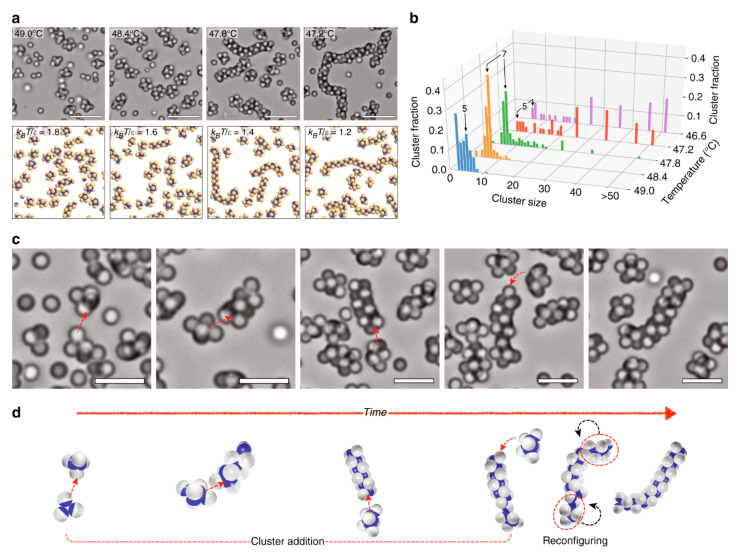
Self-assembly of Janus particles into clusters and chains at a patch ratio of 0.3. (**a**) Bright-field images above and simulation plots below show self-assembled structures of Janus particles at different temperatures; (**b**) Proportion in clusters of different sizes as a function of temperature. For cluster sizes greater than 20, histogram bars integrate over ten or more bins: 21–30, 31–40, 41–50, and >50. (**c**) Snapshots and (**d**) Figures show the formation of chain structures by collective polymerization through cluster addition and reconfiguration. Red arrow represents the formation of chain structure. Reproduced with permission from Oh et al. (2019), ©Nature Communication, 2019 [63].

**Figure 8 pharmaceutics-15-00423-f008:**
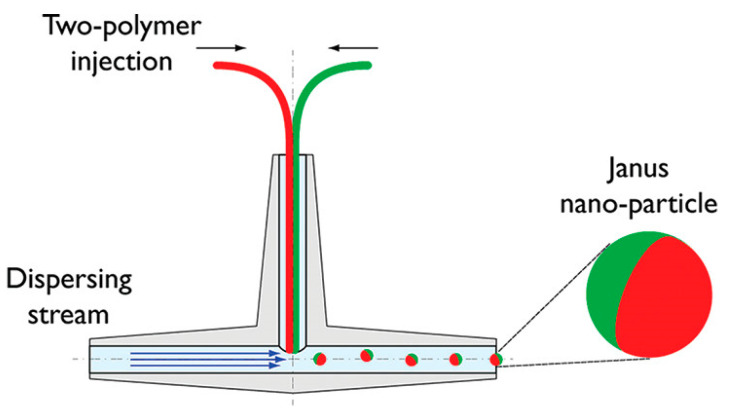
Schematic of the one-step fluidic nanoprecipitation system to fabricate Janus particles. Reproduced with permission from [64], © American Chemical Society, 2012.

**Figure 9 pharmaceutics-15-00423-f009:**
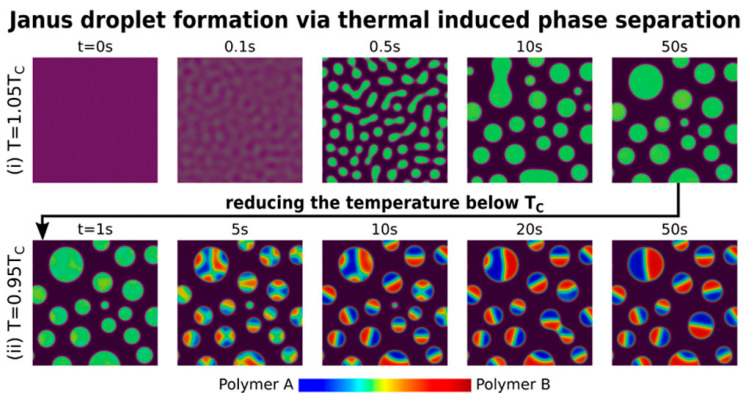
Schemes showing the thermally induced phase separation for the formation of Janus droplets. Reproduced with permission from Zhang et al. (2022), ©American Chemical Society (ACS), 2022 [70].

**Figure 10 pharmaceutics-15-00423-f010:**
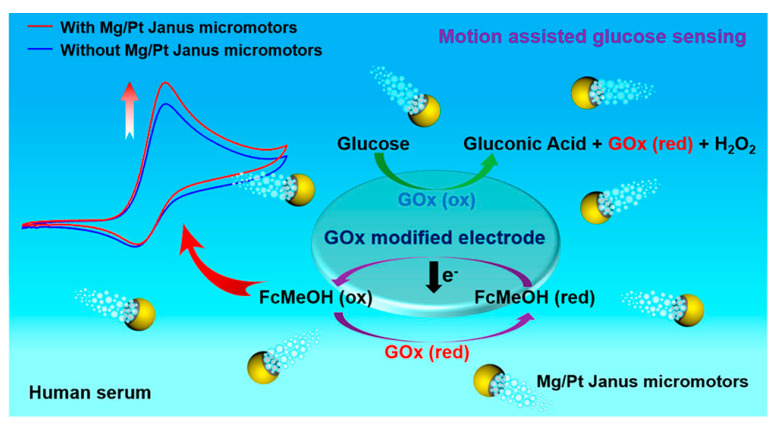
Schematic Representation of magnesium/platinum-based Janus micromotor assisted glucose biosensor in human serum using screen printed electrode. Reproduced with permission from Kong et al. (2019), ©American chemical society (ACS), 2019 [105].

**Figure 11 pharmaceutics-15-00423-f011:**
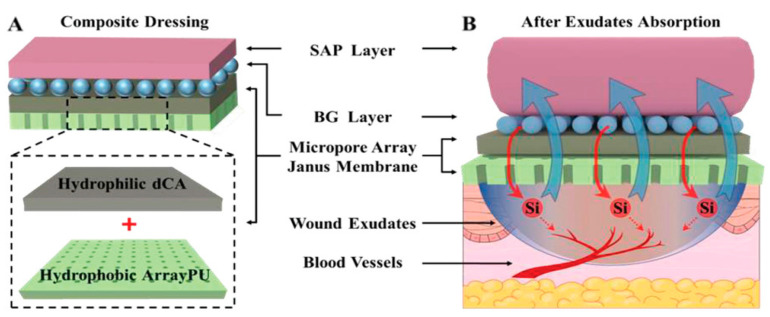
Illustrations of (**A**) composite dressing structure with modified Janus membrane and (**B**) absorption of massive wound exudates from the wound bed. SAP—sodium polyacrylate superabsorbent particles; BG—silicate bioglass; PU—polyurethane; dCA—deacetylated cellulose acetate layer. Reprinted with permission from [116], © Wiley, 2020.

**Figure 12 pharmaceutics-15-00423-f012:**
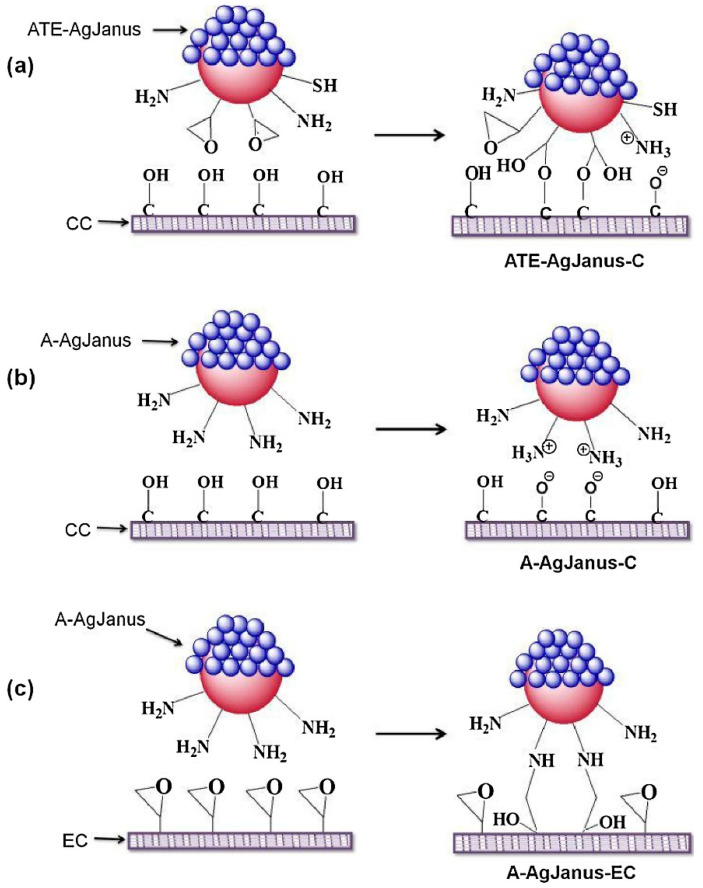
Silver-silicon dioxide Janus nanoparticles with distinct functional groups, such as amine, thiol, and epoxy synthesized via the Pickering emulsion method. (**a**) Mechanistic attachment of amine-thiol-epoxy functional ATE-silver (Ag) Janus on cotton fabric, (**b**) amine functionalized silver Janus interacts with cotton fabric via electrostatic attraction and (**c**) attachment of amine functionalized silver Janus on epoxy functionalized cotton fabric. Reproduced with permission from [122], ©Elsevier (2018).

**Figure 13 pharmaceutics-15-00423-f013:**
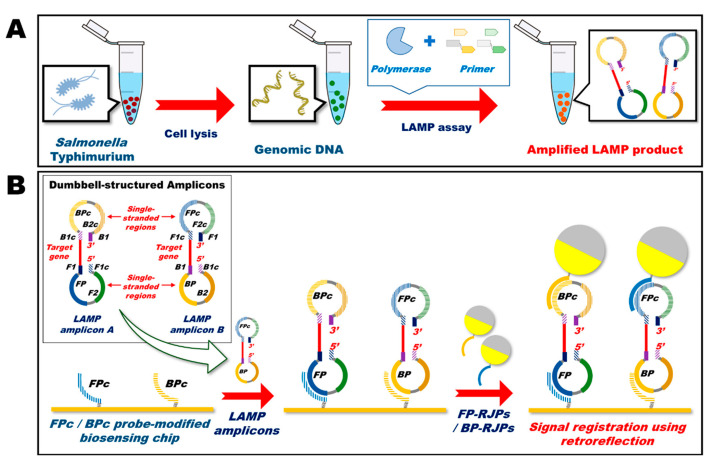
Schematics of optical biosensing platform based on the LAMP assay. (**A**) Synthesis of amplified LAMP product and (**B**) Preparation of non-spectroscopic signaling probe. Reproduced with permission from Chun et al. (2018), ©American Chemical Society, 2018.

**Table 1 pharmaceutics-15-00423-t001:** Generation of Janus particles via distinct fabrication techniques for various applications.

Janus Particles	Synthesis Technique	Formation and Characteristic Features	Application	References
Hexanethiolate-capped Janus gold (AuC6) nanoparticles	Masking	Langmuir method and gas-liquid interface approaches were used to form the amphiphilic Janus particles. Hydrophobic alkanethiolate gold nanoparticles were added to water to form a monolayer of gold nanoparticles. Air pressure was raised to force the gold hydrophobic monolayer into the water media, resulting in a reduced contact angle. Hydrophilic thiol, 3-mercaptopropane-1, 2-diol were added to competitively replace the hydrophobic thiols, leading to the successful formation of Janus particles.	Generation of sophisticated and functional nanostructures via careful selection of chemical ligands.	[22]
“Mushroom-like” Janus poly(methyl methacrylate) (PMMA)/poly(styrene-2-(2-bromoisobutyryloxy)ethyl methacrylate)-graft-poly(2-(dimethylamino)ethyl methacrylate) (PDM) particles	Phase separation	A mushroom-shaped, micron-sized, and monodispersed PMMA/P(S-BIEM)-graft-poly (DM) Janus particles were produced using a 2-step method. Firstly, the spherical PMMA/P(S-BIEM) Janus particles were synthesized via internal phase separation triggered by solvent evaporation. Subsequently, surface-initiated polymerization was used to form the Janus particles. The Janus particles were pH-dependent and able to induce volume phase transition reversibly, allowing switching between hydrophobic and amphiphilic properties due to the nature of DM.	They can be applied as particulate surfactants for o/w emulsion due to their dual stimuli-responsive behavior.	[23,24]
Pyrene labelled-magnetic nanoparticles (MNPs) linked-poly(styrene-block-allyl alcohol) (PS-b-PAA) Janus nanoparticles	Phase separation	Nano-sized Janus particles with spatially separated functionalities. A simple and scalable method was developed based on microemulsion and controlled phase separation. Hydrophobic magnetic nanoparticles and amphiphilic poly(styrene-block-allyl alcohol) copolymer were used to form the Janus particles with the magnetic field-responsive property.	Magnetic field modulated imaging application and magnetolytic therapy. The multifunctional features show promising potential for nanomedical, spintronic, and electronic fields.	[25]
Poly (*tert*-butoxystyrene)-block-polybutadiene-block-poly(*tert*-butyl methacrylate) (tSBT) polymeric Janus particles	Controlled phase transitions	Three different shapes of Janus particles (cylinder, sheets, and ribbons) were synthesized via controlled phase transition using a specific tri-block terpolymer called poly(*tert*-butoxystyrene)-block-polybutadiene-block-poly(*tert*-butyl methacrylate) (tSBT). The Janus particles have the potential to offer more functionalities as they are water soluble at a high pH.	For tuning of bulk morphologies to form soft Janus particles with nano-scaled dimensions and different non-spherical shapes from a single triblock terpolymer. The technique improves water solubility and the stimuli responsiveness of Janus particles, enabling tailored functionalities in aqueous media.	[26]
Magnetic Janus particles	Solvent evaporation-induced phase separation and microfluidic technique	Fabrication of Janus particles with controlled size desired morphological features, and size distribution by controlling the volume ratio of polymers as well as the non-equilibrium aspects of the phase separation process. The technique is suitable for fabricating colloids with varying morphologies and functions.	Facilitates the use of a single emulsification step to create a versatile microfluidic technology that can generate various structural compositions of Janus particles.	[27]
Amphiphilic Janus particles	Single-emulsion polymerization method	Synthesis process using the monomers of soybean oil-epoxidized acrylate (SBOEA) via single-emulsion droplets, butyl acetate, ethyl cellulose, and initiators. Glass-silicon microfluidic device has been utilized for the large-scale generation of uniform EC/soybean oil polymer Janus particles	The study showed the synthesis of highly natural amphiphilic Janus particles with plant-derived materials. These Janus particles are identified to possess the ability to stabilize oil-in-water emulsions under flowing conditions.	[28]

**Table 2 pharmaceutics-15-00423-t002:** Janus particles as an effective drug-delivery system for the treatment of various diseases.

Janus Particles	Characteristics Features and/or Performance	References
Hybrid plasmonic-magnetic and biocompatible SiO_2_-coated Ag/Fe_2_O_3_ Janus particles.	The physical characteristics and functionality of each compartment of the Janus particles demonstrated the potential for clinical applications in thermal therapy, magnetic or optical resonance imaging, and targeting drug delivery. The in vitro experimental findings indicated their capabilities as biomarkers via specific interactions with tagged HeLa and Raji cells.	[80]
Biocompatible Janus particles are made up of poly(lactic-*co*-glycolic acid) (PLGA).	PLGA Janus particles were generated via one-step fluidic-dependent nanoprecipitation, incorporating nanoprecipitation for hydrophobic and emulsion for hydrophilic drug molecules. The in vitro studies demonstrated that PLGA Janus nanoparticles were capable of encapsulating multiple pharmaceutical ingredients of different solubilities, such as paclitaxel (hydrophobic) and doxorubicin (hydrophilic) drug molecules, for drug delivery.	[64]
‘Handbags-type’ Janus particles with 2 separate oil and aqueous cores surrounded by a phospholipid.	This positively charged Janus particle is comprised of medium-chain triglycerides, lecithin, stearyl amine, and poloxamer 188. The Janus particles showed effectiveness in carrying both hydrophobic and hydrophilic pharmaceutical ingredients according to the in vitro examinations. It was developed for DNA condensation and delivery transfection.	[81]
A two-in-one micelle-plex (Janus).	Both in vitro and in vivo studies showed effective co-delivery of both siRNA and paclitaxel drug molecules to the same cancer site. A triblock copolymer poly(ethylene glycol)-*b*-poly(ε-caprolactone)-*b*-poly(2-aminoethyl ethylene phosphate) was used to develop the Janus particles. The findings showed promising results for the combination of SiRNA-based treatment and chemotherapy to trigger synergetic effects.	[82]
Janus nanoparticles with ice-cream coned shape.	The Janus nanoparticles were synthesized from FDA-approved polymers such as PLGA and polyvinylpyrrolidone, and GRAS-stated lipids such as glycerol behenate and glycerol di-stearate. The generation steps included emulsification, solvent evaporation, and phase separation. The in vitro research findings indicated that PLGA/glycerol di-stearate Janus nanoparticles were efficient in carrying both curcumin and doxorubicin to target and treat lung tumors in vitro using A549 human lung tumor cells. No gene or cytotoxic effects were observed. In vivo study showed the accumulation of Janus nanoparticles in the lungs of mice for at least 24 h after nasal delivery by inhalation, indicating sustained of the drug ingredient.	[83]
Amphiphilic Janus dendrimers.	Amphiphilic Janus dendrimers were fabricated with one polar and one non-polar compartment for effective bone-targeted drug delivery. The encapsulation efficiency of naproxen within the Janus dendrimers was 28-fold higher compared to native naproxen. The Janus dendrimers exhibited a good binding rate of >80% towards the targeting of bone hydroxyapatite with low cytotoxicity in the in vitro study.	[84]
Janus particles with differentially degradable compartments.	The Janus particles were made up of poly(acrylamide-*co*-acrylic acid) and poly(ethylene oxide) (PEO) at one side, and poly(acrylamide-*co*-acrylic acid) and dextran at another side via electrohydrodynamic co-jetting and controlled crosslinking. Both sides demonstrated to have different degradation kinetics and pH-stimulated degradation, allowing a more effective oral drug delivery to occur, based on in vitro analysis.	[85]
Janus particles for theranostic application.	The Janus particle was synthesized from a hydrophilic PEI hydrogel-based release compartment and hydrophobic PLGA imaging compartment. In vitro studies showed that it is a bi-compartmental nanoparticle carrying both diagnostic and therapeutic agents, offering multiple functionalities for biomedical applications.	[86]

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
