# Peer review of "Development of Janus Particles as Potential Drug Delivery Systems for Diabetes Treatment and Antimicrobial Applications"

_pharmaceutics, 2023, doi:10.3390/pharmaceutics15020423_

Round 1

Reviewer 1 Report

ABSTRACT

1.      Please recast this sentence and please replace ‘’material to improve’’ with that could Improve in this sentence ‘’Janus particles have emerged as a novel and smart material to improve pharmaceutical 13 formulation, drug delivery, and theranostics’’

Abstract section

2.      I therefore suggest that the abstract should be rewritten in the following pattern to illuminate on key finding from this study.

a.       Brief introduction

b.      Aim and objectives

c.       Key and most important results

d.      Conclusion and contribution to knowledge.

Introduction

The authors needs to clearly states the aims and objective of this study very clearly. You need to start this in a new paragraph. You start like this ‘’Therefore, this study aims to’’

Discussion section

1.      The author need to relate the results obtained during this study with relevant discussion and compare the results obtained to previously results from other researchers

2.      The manuscript is based on a very good concept methodologically executed but poorly written. The methods failed to align with the result with the discussion. Authors need to surrender this paper to serious editorial review by an English expert or language skilled colleague. This would illuminate the manuscript and makes it more comprehensive.

Author Response

Respected reviewer, 

The authors would like to thank the reviewer for his/her valuable comments to improve the quality and standard of this manuscript.

Regards, 

Prof. Michael K. Danquah

Reviewer 2 Report

The present review deals with an interesting and attractive topic in the field of material science. The Janus particles, as one of the novel forms of drug delivery vehicles, biosensors, theranostic agents, etc. are gaining growing attention from the research community. As this is a review type of article, the authors have covered all important subjects regarding development/production, structural and functional characteristics, and some interesting examples in biomedical applications. As a special part of biomedical applications, the authors summarized the present results in the fields of diabetes treatment and antimicrobial activity. In the final part, perspective and challenges are addressed.

The manuscript is well-organized and written. The references are meaningful and informative. The authors tried to emphasize crucial and important parameters that could determine the application potential of Janus particles. Overall, good review approach. Conversely a few minor issues/suggestions could be addressed before acceptance. 

The first impression from the title is that the review is focused on the development of Janus particles intended only for diabetes treatment and antimicrobial applications. Based that the authors covered general principles, synthesis approaches, important parameters and properties of these particles, and some other examples of applications, it seems that it would be more appropriate to reformulate the title. Regarding this issue, the part that refers to antimicrobial applications is significantly shorter than other sections. Although the authors stated that this topic has not been investigated a lot, it could be expanded with additional results (maybe about bacteria sensing),  another image (if it’s possible), or some other comments. 

Figure 5. is informative, but it is also given in the other review article published in 2019. (ref. 39). If it is possible, authors could try to replace it with another image?

In the Prospect section (7.1) reference is missing for the sentence that finished in line 518 (For example, certain commercially available bilayer drug tablets are manufactured in the Janus dosage form to encapsulate and deliver two different or incompatible active pharmaceutical ingredients (APIs) for different therapeutic effects).

English is very good, but there are some small typing errors as in line 318.

It seems that affiliation 5 is not properly assigned to an adequate author.

Author Response

(The authors gave the same response as above.)
